# Dual U-Net-Based Conditional Generative Adversarial Network for Blood Vessel Segmentation with Reduced Cerebral MR Training Volumes

**DOI:** 10.3390/mi13060823

**Published:** 2022-05-25

**Authors:** Oliver J. Quintana-Quintana, Alejandro De León-Cuevas, Arturo González-Gutiérrez, Efrén Gorrostieta-Hurtado, Saúl Tovar-Arriaga

**Affiliations:** 1Faculty of Engineering, Autonomous University of Querétaro, Querétaro 76010, Mexico; oliverqquintana@gmail.com (O.J.Q.-Q.); aglez@uaq.mx (A.G.-G.); efrengorrostieta@gmail.com (E.G.-H.); 2National Autonomous University of Mexico (UNAM) Juriquilla Campus, Queretaro 76230, Mexico; a.deleon.cuevas@comunidad.unam.mx

**Keywords:** brain blood vessels segmentation, conditional generative adversarial network, MRI segmentation, residual U-Net, time-of-flight magnetic resonance angiography

## Abstract

Segmenting vessels in brain images is a critical step for many medical interventions and diagnoses of illnesses. Recent advances in artificial intelligence provide better models, achieving a human-like level of expertise in many tasks. In this paper, we present a new approach to segment Time-of-Flight Magnetic Resonance Angiography (TOF-MRA) images, relying on fewer training samples than state-of-the-art methods. We propose a conditional generative adversarial network with an adapted generator based on a concatenated U-Net with a residual U-Net architecture (UUr-cGAN) to carry out blood vessel segmentation in TOF-MRA images, relying on data augmentation to diminish the drawback of having few volumes at disposal for training the model, while preventing overfitting by using regularization techniques. The proposed model achieves 89.52% precision and 87.23% in Dice score on average from the cross-validated experiment for brain blood vessel segmentation tasks, which is similar to other state-of-the-art methods while using considerably fewer training samples. UUr-cGAN extracts important features from small datasets while preventing overfitting compared to other CNN-based methods and still achieve a relatively good performance in image segmentation tasks such as brain blood vessels from TOF-MRA.

## 1. Introduction

Brain vessel segmentation locates blood vessels in brain image volumes that can be processed to generate 3D models of the patient’s brain to visualize it for diagnosis or before any intervention. Segmentation is usually performed by hand to each slice into the volume, a task that is quite time-consuming and relies entirely on physician expertise, where misclassification can lead to severe consequences for the patient. Given this situation, it is important to develop intelligent systems that can perform vessel segmentation without the drawbacks of time consumption of experts and human error.

Brain imaging and blood vessel segmentation help neuroradiologists in trajectory planning for minimally invasive surgery [1] in detecting brain diseases, such as aneurysms, strokes, and electrode trajectories for deep brain stimulation [2,3,4,5]. The main challenges of segmenting MRI images are normally the size in memory they occupy, the complexity and time in obtaining them, the absence of large labeled datasets for training machine learning algorithms, and the high noise they present, making this a very challenging task to carry out.

Convolutional Neural Networks are among the most successful methods for image segmentation. They have gained popularity due to their advantages, such as higher Dice scores and increasing in the cloud training possibilities. They also provide an increasing corpus of techniques that can be used for different applications. Different architectures of deep convolutional neural networks, with a relatively small number of layers combined with fully connected layers to classify each pixel in a 2D medical image between vessels and non-vessels, have been proposed by [6,7]. Due to the nature of image acquisition techniques, slices are commonly stacked into 3D volumes, where adjacent images share some similarities. This property has been exploited by Chen et al. [8] to propagate information between layers of the stack for vessel segmentation, achieving a Dice score of 73.71%. Teikari et al. [9] developed a hybrid 2D-3D convolutional neural network to segment brain blood vessels from two-photon vasculature microscopy stacks using a CPU as an accelerator framework, proving that deep neural networks can be optimized enough to be trained without a GPU. Taking into consideration the signal variability given by blood flow, Kandil et al. [10] divide their MRA volumes into two parts, above and below the Circle of Willis (CoW), and both are fed into their 3D CNN architecture, with an 84.37% Dice score as a result. Zhao et al. [11] developed a framework that extracts MRA volume structures as a preprocessing step. A fully connected neural network acts as a classifier that takes as input several properties of candidate structures and obtains the probability of being a blood vessel.

Some of the most successful techniques in machine learning are Transfer Learning (TL), Data Augmentation, and Generative Adversarial Neural Networks (GAN’s). Many authors have tried to use those techniques looking for an improvement in TOC-MRI segmenting tasks. The work presented by Tetteh et al. [12] relies on TL for three main tasks, location of the centerline and bifurcation points in blood vessels, as well as segmentation, using a synthetic dataset to pre-train their 3D CNN, validating on MRA images of human brains and rats, where they achieved 86.68% Dice score. Using data augmentation, Zhang et al. [13] apply a reflection transformation to the dataset to generate more samples needed, given the disadvantage of not having the necessary amount of training data. In addition to this, they add Gaussian noise to the samples to make the network invariant to absolute intensities. They proposed a hybrid loss function to handle class imbalance between vessels and non-vessels regions with dense connections to connect feature maps within the CNN architecture. As a result, they present a 75.6% Dice score for vessel segmentation in susceptibility-weighted images (SWIs). Image generation models such as generative adversarial networks (GAN) [14] are gaining popularity for various tasks, where especially conditional GANs [15] have been used to generate images from another one, adding random noise, useful for style transfer. Image segmentation can be seen as a style transfer problem. This point of view is used by some authors [16,17,18,19,20,21] for different applications, including the segmentation of organs, blood vessels, and tumors. 

Encoder-decoder architectures have shown increased performance in different segmentation tasks. In medical imaging, the U-Net model proposed by Ronneberger et al. [22] has established an inflection point in the medical community, demonstrating that a deep convolutional neural network can be trained with small datasets, which is often necessary given the time-demanding task of obtaining labeled images from specialists, and still achieve competitive results. Several U-net-like architectures have been proposed to attend to this issue for different tasks in various applications, including segmentation of cerebral blood vessels, brain tumors, ischemic-stroke lesions, aneurysms, and skin lesions [23,24,25,26,27,28,29]. Concatenations of U-Nets have also been demonstrated to help improve the segmenting of smaller blood vessels in retina fundus images [30].

In the present work, we will focus on brain vessel segmentation for Time-of-Flight (TOF) MRA, using a conditional generative adversarial network with an adapted generator based on the concatenation of a U-Net [22] and a residual U-Net [31] architecture. The model is trained on a small set of samples compared to other state-of-the-art methods, given the limited availability of labeled data, while achieving similar evaluation metrics results. Manually annotated data is very time-consuming to elaborate on due to the number of slices within each MRA volume. Additionally, an expert is required to perform such a task, as other cerebral regions may interfere, causing wrong-labeled regions that could threaten the patient’s health. Therefore, approaches that rely on few volumes are essential to solve a problem where a big challenge is the lack of available data while reducing the time needed for an expert to visualize cerebral blood vessels’ segmentation.

## 2. Materials and Methods

A.Data

The 3D TOF-MRA images used in this work were provided by the Institute of Neurobiology of the National Autonomous University of Mexico (UNAM), which are composed of four volumes from four patients (3 females aged 55, 41, and 33 years old and a 23 year-old male) obtained by a Phillips Achieva 3T system (Phillips Healthcare, Amsterdam, Netherlands). Each volume contains 200 slices with a resolution of 560 × 560 pixels, with a separation of 0.5 mm between each one. Figure 1 shows an example of the dataset. Additionally, a segmentation map for each slice is provided, annotated by hand, for model training purposes.

B.Image preprocessing

To reduce the MRA volumes’ background noise, a mask is generated to preserve the Region of Interest (ROI) for each slice. First, all pixels with intensity below the threshold = 10 are reduced to 0, followed by a morphological closure with kernel size k = 9 for the mask generation. After this, the mask is multiplied by the original image to obtain the ROI only, without background noise, as shown in Figure 2.

C.Model Architecture

GANs are generative models that consist of a generator *G* and a discriminator *D*, with parameters $\theta^{\left(G\right)}$ and $\theta^{\left(D\right)}$, respectively. *G* learns a mapping from a random noise vector *z* to an output image *x*, represented by $x=G\left(z;\theta^{\left(G\right)}\right)$, whereas the task for *D* is to distinguish between samples from the training data and samples obtained from the generator, denoted by $D(x;\theta^{\left(D\right)})$ [14,32].

The model function V(D, G) is represented by a min-max game, where *D* is trained to maximize the probability of a correct label between real and fake samples, while *G* tries to minimize it generating samples indistinguishable from the real ones, as follows:(1)$$min_{G}max_{D}V\left(D,G\right)=E_{x∼P_{data\left(x\right)}}\left[logD\left(x\right)\right]+E_{z∼P_{z}\left(z\right)}\left[\log\left(1-D\left(G\left(z\right)\right)\right)\right]$$

The purpose of the model is that the discriminator learns the properties from training samples, denoted as true examples, and the ones from the generator denoted as fake. From the discriminator classification, the generator is forced to generate better samples indistinguishable from the training data.

Conditional generative adversarial networks add the property to conventional GANs of having a sample y alongside the input z as extra information to condition the model to generate samples based on the input, a useful feature for image segmentation, where the output sample is similar to the input, and a completely different image is not required to be generated. The objective function for this model follows the next equation:(2)$$min_{G}max_{D}V\left(D,G\right)=E_{x∼P_{data\left(x\right)}}\left[logD\left(x|y\right)\right]+E_{z∼P_{z}\left(z\right)}\left[\log\left(1-D\left(G\left(z|y\right)\right)\right)\right]$$

The proposed architecture is based on the Pix2Pix architecture [33] and shown in Figure 3, which consists of two models. First, a generator model *G* with two stages similar to [30]; an encoder and decoder sections to have a U-net- like architecture, concatenated with a similar one, but adding residual blocks [30,31]. Second, a discriminator *D* model based on the encoder section of the first part of the generator.

The architecture takes a single-channel image or slice from the 3D MRA volume with a resolution of 256 × 256 pixels and noise distribution function as dropout [34] that is fed to the generator. The discriminator’s output is a 16 × 16 feature map, as [33] suggests.

For the encoder from the first section of *G*, four 3 × 3 convolutions are implemented with the same padding, the first one with stride 2, following instance normalization and leakyReLU activation functions. The decoder section performs four 3 × 3 transposed convolutions to retrieve feature map size, followed by another 3 × 3 convolution, with instance normalization and leakyReLU activation functions.

The second section of the generator is very similar to the one described before, with the only difference of replacing the convolutions after the dimension reduction of the feature map by a residual block. The same difference is noted in the decoder part after each transposed convolution. Tanh is used as an activation function for the last convolutional layer of the generator model.

The residual blocks are based on [30] and represented by Equation (3) and Figure 4, where each one takes as input an n-dimensional feature map FM. First, a 3 × 3 convolution is performed with the same padding, followed by instance normalization, LeakyReLU activation function, another 3 × 3 convolution, and instance normalization. The output is combined with the *FM* as a sum operation, and a final LeakyReLU activation function is then used.
(3)FM(x)=F(x)+x

The discriminator model is based on the first section of the encoder generator with filters of size 4 × 4, keeping the same operations and adding a final single-filter convolution, with the same padding to generate a 16 × 16 output. LeakyReLU is used as an activation function for all layers except for the last, which is replaced by a sigmoid function.

D.Training Strategy

The final architecture is assembled as Figure 5 shows for both models, *G*, and *D*, within the cGAN model, a generator followed by a discriminator model, having as output a binary array containing whether a generated image is classified as real or fake. Based on this classification, the generator model is trained while the discriminator weights remain non-trainable. Due to the concatenation of two U-Net-like models for *G*, the total number of parameters is 65 million, which is almost equivalent to twice the parameters in a conventional U-net, whereas the parameters for *D* are 6.9 million.

The discriminator’s learning procedure trains on real and fake image batches without affecting the generator weights.

Due to their nature, various medical images contain imbalanced classes. Training machine learning algorithms with imbalanced classes is an area of research interest [35,36]. In this work, two loss functions are used, one for each model within the architecture. *L*1 loss is used for the generator model altogether the cGAN loss, whereas binary cross-entropy loss function is implemented for the discriminator model.

The procedure for training the models is based on [14], where a step for gradient descent on the discriminator model is performed, followed by one for the generator. The discriminator is trained as a conventional CNN model, while the generator makes use of the loss from the inference of the discriminator to be trained. The objective function is shown in Equation (4). A combination of two-loss functions is implemented as in [33] for *G*, with *L*1 loss as described by Equation (5). Initial hyper-parameter selection is based on [33] as well.
(4)$$g=\arg min_{G}max_{D}{𝓛}_{cGAN}\left(D,G\right)+\lambda {𝓛}_{L1}\left(G\right)$$
where:(5)$${𝓛}_{L1}\left(G\right)=E_{x,y,z}\|y-G\left(x,z)\right\|$$

E.Implementation Details

Each slice in the MRA volumes is resized from 560 × 560 pixels to 256 × 256 pixels to reduce the memory consumption while training the model.

Data augmentation is implemented for each slice in the training set in order to generate more samples by two different techniques. The first one consists of flipping each sample with a mirror transformation, whereas the second one is patch extraction, which applies random zoom (between the original and twice the size of the sample) into random areas of the original data.

Different values for λ were tested. After several experiments, a proportion of 75 to 1 in favor of *L*1 loss was chosen, as the best results for segmentation from the model were achieved. A random noise feature map is commonly added to the input map for the generator model. As a replacement, dropout is implemented in the bottleneck section of the model. Similarly, the discriminator also uses dropout at the final layer of the model. For each model, the probability is equal to 0.1.

The generator model is trained using Adam optimizer [37], initial learning rate α=0.0002 with an exponential decay rate of 0.9, $\beta_{1}=0.9$ and $\beta_{2}=0.999$, whereas the discriminator uses RMSProp [38] for training, as it leads to better stability at minimizing its cost function, with the same learning rate as the generator and momentum =0.9. Besides, the optimization pace for *D* is divided by 2 with respect to *G* [33].

Instance normalization is implemented for better stability and improved performance in the learning procedure, as [39] suggests for image generation models.

The proposed model is trained on a computer with Ubuntu 18.04, 56 GB of RAM, and an Nvidia K80 GPU. Python 3.6, TensorFlow 2.3.0 with its Keras library, is used to build the architecture.

## 3. Results

The similarity between segmented prediction and target is evaluated using different metrics, such as Dice score (DSC), accuracy, precision, sensitivity, and specificity, given by the following expressions, where true positive (*TP*) and true negative refer to the pixels properly classified as blood vessel and background by the evaluated models, respectively, whereas the false positive (*FP*) and false negative (*FN*) indicate misclassified pixels; background predicted as blood vessel and blood vessel predicted as background, respectively.
(6)$$Dice\;Score=\frac{2TP}{2TP+FP+FN}$$
(7)$$Precision=\frac{TP}{TP+FP}$$
(8)$$Sensitivity=\frac{TP}{TP+FN}$$
(9)$$Specificity=\frac{TN}{TN+FP}$$

A.MRA Images

The dataset is split into subsets, each used as validation data for a k-Fold validation, with *k* = 4. Therefore, 4 models are trained using different validation subsets for each iteration, exactly 3 complete volumes for training and 1 for validation at each step. Table 1 summarizes the evaluation made for each model and the average for the metrics evaluated on the validation volume for the corresponding iteration *k* of the *k*-Fold experiment, and Figure 6 shows different slices from the MRA volumes, ground truth segmentation map, segmentation prediction, and error map.

In Table 2, metrics evaluated for different approaches are shown. U-Net, concatenated U-Nets, and Pix2Pix, the models from the proposed one, follow the same methodology for k-Fold. The CNN models were trained with a similar selection of hyper-parameters, based on the work of [30,31], using dropout to diminish overfitting, with a probability of 0.2 between layers, and early stop after training for 10 epochs with no Dice Score improvement for the test set. Similarly, the Pix2Pix model trained for this work uses the hyper-parameter selection proposed by the authors in [33]. For all methods, an exponential learning rate schedule was added with a decay rate of 0.96 and a batch size of 10. A 3D visualization for a test volume is shown in Figure 7 for each model, and in Figure 8, the error maps for each of the evaluated models can be visualized, where the maps are obtained from the absolute difference between the ground truth and the predicted volumes.

It is important to mention that a direct comparison cannot be made between methods due to the variance of the datasets used for each one. However, a reference is useful, as shown in Table 3.

B.Additional Experiment Using Microscopy Images

In addition to the evaluated MRA dataset, we tested the same experiment as described across this work but using an additional dataset publicly available, based on two-photon microscopy cerebral images [9], with its corresponding segmentation map for blood vessels. The objective for this is to prove that our model is able to work with relatively good performance, in comparison to the other tested models, while being trained on a different image modality aside from the MRA images.

Each slice for each volume was preprocessed to enhance the ROI. This was performed following the CLAHE method, proposed by [40], where background vessels can be better identified with a higher value, with the drawback of adding noise to the image, as shown in Figure 9. Nevertheless, this led to better performance for our experiments, with the aid of data augmentation given the small number of samples available in the dataset.

For this experiment, a 4-Fold method was implemented in order to split the 12 volumes in the dataset into 9 and 3 volumes for training and validation, respectively for each iteration for the 4-Fold, cycling them each time, so each volume is used only once for validation. The results for this experiment, evaluating the models presented in this work, are shown in Table 4, and a sample of the segmentation maps in Figure 10. Also, the hyperparameter selection for all models remains the same as described in Section 2 and Section 3.

The evaluation of this dataset is useful to provide additional information on how our proposed model behaves while not using MRA images, which is our main goal, for blood vessel segmentation and to demonstrate that it could be helpful in a variety of segmentation tasks. Further hyper-parameter tuning can be applied to the models tested, but we followed the same training approach discussed in this work to make a fair comparison for all data and models.

## 4. Discussion

According to our experiments, the U-Net model tends to overfit easily, given the small number of training samples available. Even with the implementation of different regularization techniques, it generates noise in locations around vessel structures, while some areas with similar pixel intensities are misclassified as vessels. The concatenated U-Nets model generates a segmentation volume with less noise overall but with a higher false-negative rate, therefore, showing a lower sensitivity than the single U-Net. However, it obtains higher precision and a better similarity, and therefore a higher Dice score. The addition of the second residual U-Net provides a better identification and posterior segmentation of small regions of interest for such a task, preserving key elements like the thinnest blood vessels. 

The Pix2Pix model addresses the overfitting problem improving the generalization performance. Therefore, we considered deploying the concatenated U-Nets, which performs a higher classification accuracy of small vessels in high unbalance classes with less training time into a cGAN model; instead of using a single U-Net as the Pix2Pix does. We show that, utilizing the concatenated U-Nets as the cGAN generator, it is possible to achieve the advantages of both architectures, eliminating the overfitting problem and improving the Dice score for unbalanced classes. Furthermore, if we compare the ground truth of small vessel segmentation, Figure 7a, we can easily see that the proposed model achieves better definition and less noise to each of the models.

cGAN models tend to perform well in image generation tasks, such as image segmentation, due to the combination of two-loss functions to minimize within the model. First, the *G* model loss is similar to any other CNN-based model loss, like the distance between prediction and target segmentation, for instance. Second, the *D* model’s objective is to distinguish key elements within both the real samples and the segmentation predictions to detect errors made by *G*. This, later on, translates into the second loss that is used, in addition to the first mentioned, to penalize prediction errors.

In Table 3, we can see that the proposed UUr-cGAN model achieves a higher Dice score than all the state-of-the-art TOF-MRA segmentation models, but the one proposed by Livne et al. [23] (which is based on a single U-Net model). It is worth mentioning that in their study, they used 66 volumes compared to the 4 volumes used in ours.

One limitation of the proposed method, as is common with deep learning models, is the high number of trainable parameters of the cGAN architecture. The whole model needs to be trained, and as discussed in Section 2C, the architecture is equivalent to having two different convolutional models (generator and discriminator) for which their respective parameters have to be trained, one step at a time for one and the other in the direction of the gradient descent for a high number of epochs, resulting in a higher requirement of resources in comparison to training a conventional convolutional network and thus, in our experiments, the associated training time can take more than 6 h for each k-fold for our dataset. The advantage is that for inference, only the trained generator model is required and the discriminator can be dismissed, as it is only required in order to update the parameters of the generator at the training stage.

## 5. Conclusions

Since the concatenation of U-Nets proved to be effective in other imaging modalities to segment blood vessels, including the thinnest ones, our hypothesis that their use as a generator in a cGAN was positively tested. As we can see, our method achieves better performance than most approaches in TOF-MRI segmentation. It is important to mention that our model can consistently segment small vessels better than other approaches, as we can see in Figure 7. Our proposed UUr-cGAN model reaches 87.23 in Dice score for blood vessel segmentation of TOF-MR Angiographies, a score superior to most approaches, with the additional advantage of obtaining this result with a considerably smaller amount of training volumes. Even though it is not directly comparable to other approaches, due to different training data, it is a good approximation because most successful models are based on the U-Net architecture.

An additional analysis of the architecture for the generator model of our proposed method might be useful to improve even further our results for this kind of segmentation, like the integration of modern techniques for more efficient training in the deep learning field, such as attention gates and deep supervision [41]. Also, the pre-processing and data augmentation strategy can be optimized using other convolutional models as sample generators [42], which might lead to a wider variety of generated samples available for training without affecting the correct distribution of the data, while preventing overfitting more efficiently.

Small datasets are a limitation for many applications across different fields where segmentation is needed. The presented work is a plausible approximation to solve many segmentation problems in the lack of large datasets while detecting small features on the images.

## Figures and Tables

**Figure 1 micromachines-13-00823-f001:**
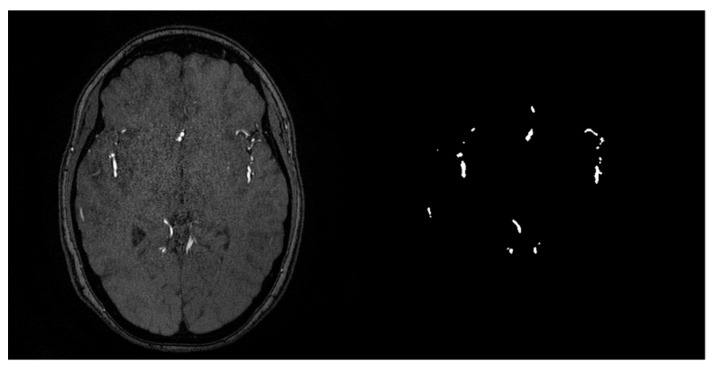
Axial view from a sample of the MRA dataset and its segmentation map.

**Figure 2 micromachines-13-00823-f002:**
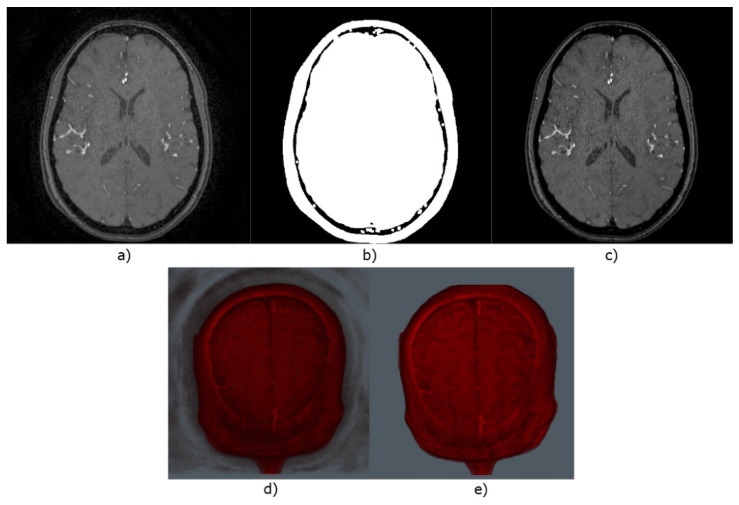
Comparison for original and preprocessed images. (**a**) Sample image from an MRA volume. (**b**) Generated mask from (**a**). (**c**) Output image from multiplying (**a**,**b**). (**d**) Sample volume without preprocessing. (**e**) Output volume from applying the corresponding mask for each image within it.

**Figure 3 micromachines-13-00823-f003:**
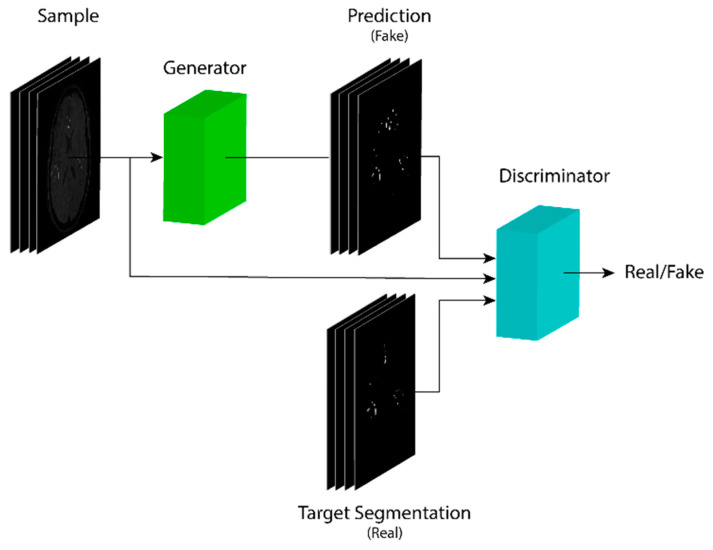
UUr-cGAN architecture. The generator model predicts a segmentation map from an input sample that is used to train the discriminator against fake samples. The discriminator model also trains for real samples based on ground-truth segmentation maps to predict the class for a given image.

**Figure 4 micromachines-13-00823-f004:**
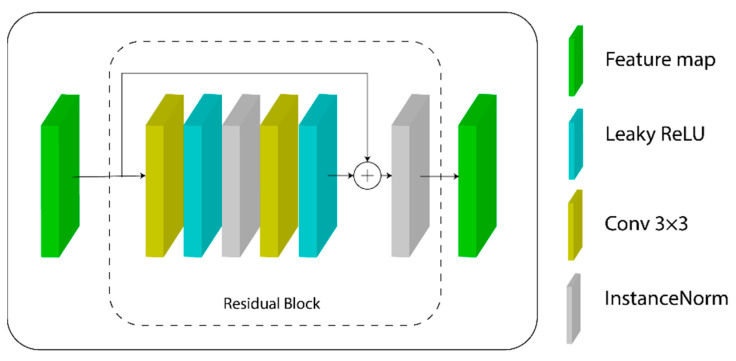
Architecture for the residual block within the second part of the generator model for the cGAN. A skip connection is used to preserve information from the block’s input feature map to the last layers, combining them as a sum operation.

**Figure 5 micromachines-13-00823-f005:**
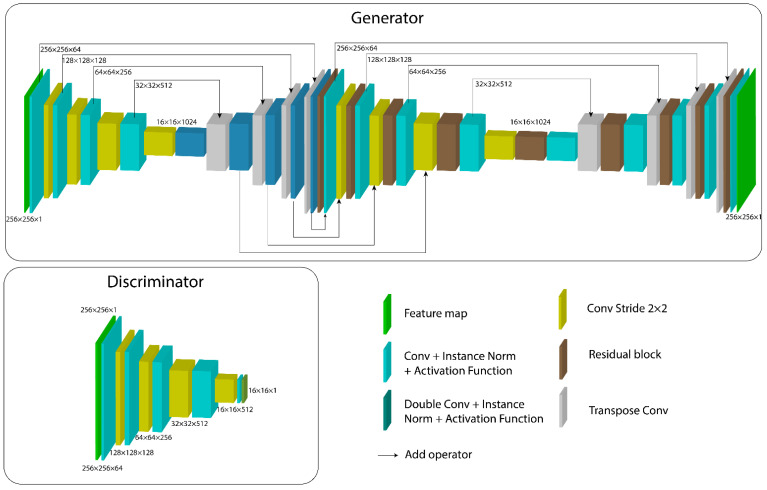
Generator and discriminator models based on [22,30,31,32,33] for the proposed cGAN architecture. A concatenation of two U-Net-like models inspires the generator. The second one contains residual blocks, which helps preserve and transfer information through deeper layers of smaller details within the feature map.

**Figure 6 micromachines-13-00823-f006:**
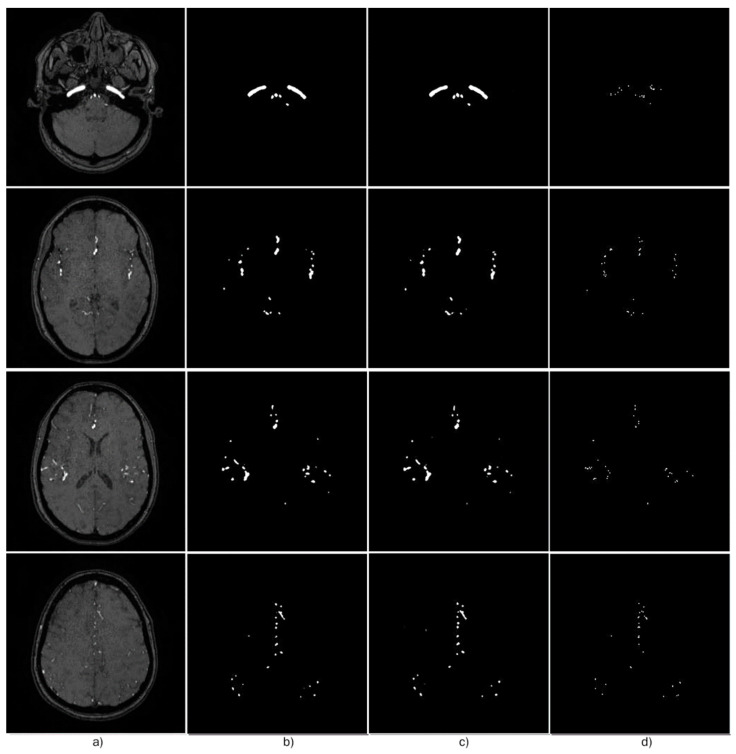
Example of vessel segmentation performed to MRA images. (**a**) MRA slices, (**b**) Segmentation target, (**c**) Prediction from the proposed model, and (**d**) Segmentation error (the difference between segmentation target and prediction).

**Figure 7 micromachines-13-00823-f007:**
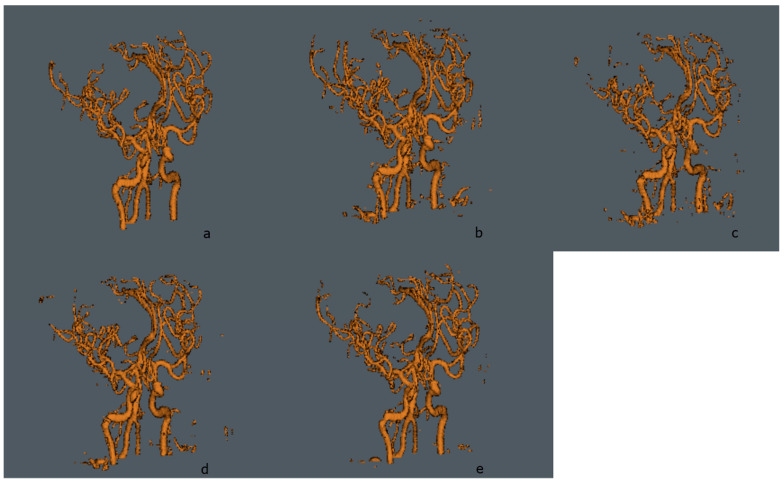
Segmentation map obtained from each one of the evaluated methods described in Table 2, using data from the first fold of the cross-validated experiment. 3D Visualizations were generated by the software Aliza Medical Imaging ©, Bonn, Germany, and DICOM Viewer for ground truth and prediction volumes. (**a**) Ground truth, (**b**) U-Net [22], (**c**) Pix2Pix [33], (**d**) Concatenated U-Net and residual U-Net [30], and (**e**) Proposed cGAN.

**Figure 8 micromachines-13-00823-f008:**
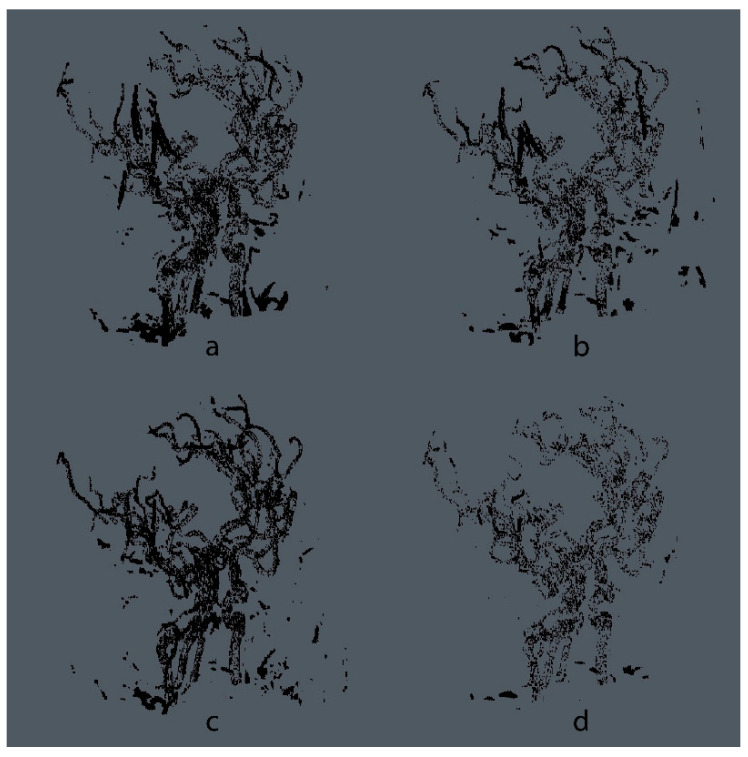
Error maps obtained from the difference between the ground truth and the predicted segmentation maps from each of the evaluated models, using the same volume shown in Figure 7. Visualizations were generated by the software Aliza Medical Imaging ©, Bonn, Germany, and DICOM Viewer. (**a**) U-Net [22], (**b**) Concatenated U-Net and residual U-Net [30], (**c**) Pix2Pix [33], and (**d**) Proposed cGAN.

**Figure 9 micromachines-13-00823-f009:**
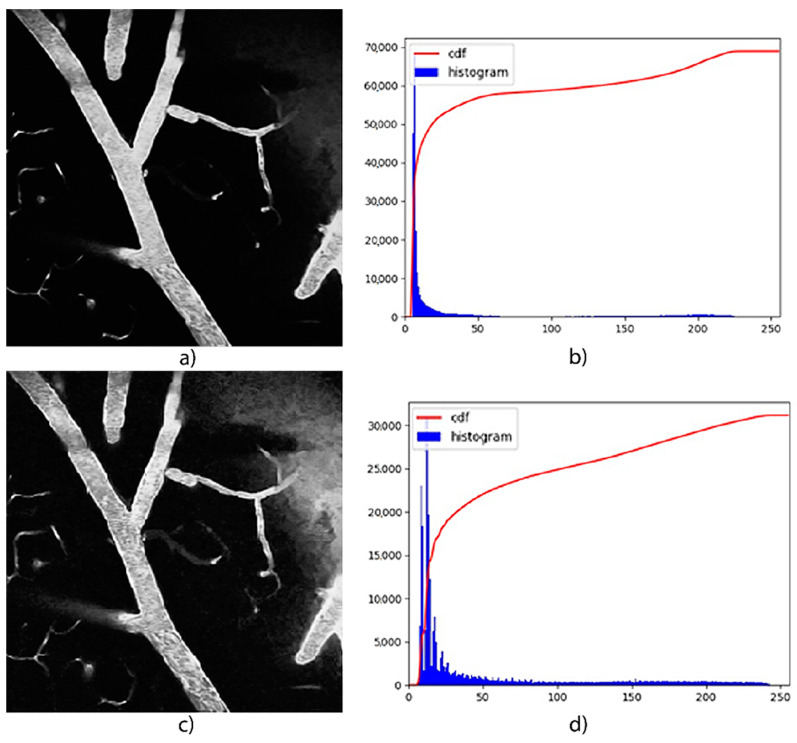
Comparison for a sample of the microscopy dataset [9] before and after histogram equalization. Image (**a**) and its histogram (**b**) correspond to the original data, whereas image (**c**) and its histogram (**d**) correspond to the enhanced by the CLAHE method [40]. Each histogram shows the distribution and the cumulative distribution function for the pixel intensity values present in its corresponding image. For the optimized image, the pixel values are slightly better distributed in comparison with the original image, which leads to a better classification of the blood vessels.

**Figure 10 micromachines-13-00823-f010:**
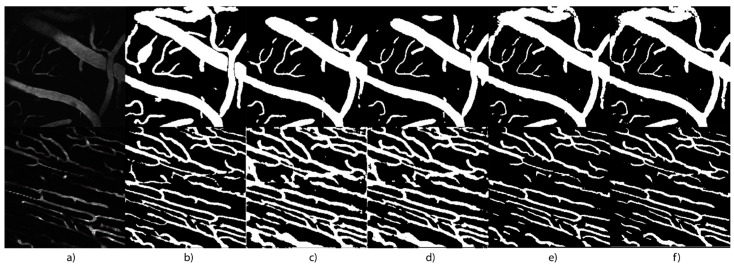
Sample of the segmentation maps for blood vessels obtained from the trained models on the microscopy dataset. (**a**) Input image. (**b**) Ground truth segmentation. (**c**) U-Net prediction. (**d**) Concatenated U-Nets. (**e**) Pix2Pix. (**f**) UUr-cGAN (proposed method).

**Table 1 micromachines-13-00823-t001:** Metrics evaluated from models obtained by each iteration of k-Fold.

k	Dice Score	Precision	Sensitivity	Specificity
1	0.8825	0.8789	0.8864	0.9996
2	0.8574	0.9064	0.8022	0.9997
3	0.8742	0.9141	0.8387	0.9997
4	0.8752	0.8817	0.8746	0.9996
Average	0.8723	0.8952	0.8504	0.9996

**Table 2 micromachines-13-00823-t002:** Average k-Fold metrics for different methods and proposed model.

Method	Dice Score	Precision	Sensitivity	Specificity
U-Net [22]	0.8371	0.7978	0.9384	0.9994
Concatenated U-Nets [30]	0.8613	0.8795	0.8633	0.9997
Pix2Pix [33]	0.8092	0.8362	0.7838	0.9996
UUr-cGAN (Proposed model)	0.8723	0.8952	0.8504	0.9996

**Table 3 micromachines-13-00823-t003:** Dice score and data availability for different vessel segmentation in MRA images alongside the proposed model.

Method	Dice Score	MRA Volumes in Dataset
Chen et al. [8]	0.7371	10
Phellan et al. [7]	0.7740	5
Tetteh et al. [12]	0.8668	40
Kandil et al. [10]	0.8437	30
Zhao et al. [11]	0.8503	30
Livne et al. [23]	0.9210	66
Proposed model	0.8723	4

**Table 4 micromachines-13-00823-t004:** Average k-Fold metrics for different methods and proposed model, for the multiphoton microscopy dataset [9].

Method	Dice Score	Precision	Sensitivity	Specificity
U-Net [22]	0.7356	0.9720	0.5917	0.9957
Concatenated U-Nets [30]	0.7704	0.9624	0.6423	0.9936
Pix2Pix [33]	0.8187	0.9772	0.7288	0.9986
UUr-cGAN (Proposed model)	0.8288	0.9567	0.7246	0.999

## Data Availability

The python implementation for the proposed model in this work can be found in the following public repository https://github.com/oliverquintana/UUr-cGAN (accessed on 17 May 2022).

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
