# Peer review of "Dual U-Net-Based Conditional Generative Adversarial Network for Blood Vessel Segmentation with Reduced Cerebral MR Training Volumes"

_micromachines, 2022, doi:10.3390/mi13060823_

Round 1
Reviewer 1 Report
The authors apply deep learning to 3D TOF-MRA images for automatic vessel segmentation. The critial limitation is that the number of volumes used in this work is too small (only four). Getting a lot more of volumetric data for the study should be needed, given that there are a variety of 3D vessel shapes and their intensity variations across patients, and there are different MRI scan parameters depending on the vendors and institutes, of which variabilities would be a concern for the performance of a deep learning algorithm.
Reviewer 2 Report
The paper proposes a conditional generative adversarial network with an adapted generator based on a concatenated U-Net with a residual U-Net architecture for blood vessel segmentation in Magnetic Resonance Angiography (MRA) images. The proposed methodology is evaluated experimentally, and the results are presented. The paper must be improved in terms of description of methodology and the presentation of results before it could be considered for publication.
Comments:
- In the introduction section discuss the main challenges of working with MRA images such as noisiness. Also, discuss in more detail the motivation for using deep learning.
- What knowledge gap has been bridged by this study? Many recent papers used the U-Net deep learning model architecture with various optimizations and modifications for biomedical imaging tasks. What is your specific innovation and contribution?
- The overview of the related works section is rather disappointing. The works are introduced without any order or structure. The style of presentation (the authors in some article did something) is poor. I suggest to group articles and discuss them in groups, while emphasizing similarities and differences of the approaches. Discuss more recent papers on brain image segmentation such as „U-net supported segmentation of ischemic-stroke-lesion from brain MRI slices“, „Automated detection of schizophrenia from brain MRI slices using optimized deep-features“, and „An efficient approach for the detection of brain tumor using fuzzy logic and U-NET CNN classification“, among others. A summary of the discussed methods presented in the table would be welcome.
- The image preprocessing is rather simplistic. Can you show that your background noise cancellation approach works? What about noise corrupting the brain image?
- Describe image augmentation in more detail with all parameter values used for the full replicability of the study.
- How did you select the hyper-parameter values (such as batch size, learning rate, etc) for the training of your deep learning model? Did you optimize? An ablation study may be required to support.
- How do you avoid/prevent overfitting during the training of neural network models? Why do you stop at 10 epochs?
- Support your experimental results section with confusion matrices and discuss.
- Evaluate the results using statistical analysis such as calculating 95% confidence limits or standard deviations of all performance values.
- What is the computational time complexity of the proposed methodology?
- Add the discussion section, and discuss the limitations of the proposed methodology.
- Some performance numbers are used without the units of measurement (the percents missing in abstract and elsewhere).
Reviewer 3 Report
General Consideration
The paper is well written, using a good English language. However, its structure could be improved in terms of sectioning. The Abstract is well written and presents a summary of the results that allows the reader to evaluate the interest compared to the paper already in this section, integrate keywords and "%" on the performances. In the introduction section there is also the literature review, it could be thought to divide it in two sections, check better the too many references without explanation (all of them are important?). The Materials and Methods section does not seem very well divided, the subsections are very unbalanced, however, with these subsections there is a correct flow of information, and the images are clear, even here the “%” is missing and there is the need to adjust Equation (1) format. The use of GANs is interesting and well explained. As for the Result section, it contains the first few lines, which should be moved either to the previous section or to a new section with implementation details; it is well written, but the definition of the dice score should be revised. The discussion section is small and could be extended to comment on the results in a critical manner and especially to compare them with the state of the art. As for the conclusions section, it explains well what has been achieved, some future developments should be included. The references section is well written and formatted. The results proposed in the appendix should be reported in the results section in a specific subsection. These additional experiments should be highlighted. It would be appropriate to give some dispersion metrics or statistical analysis given the very low number of samples which may make the method not statistically valid and robust.
Abstract
The section is well written and structured, the English is also good. It allows the reader to understand the problem being analysed and how the problem is to be solved. There is also a reference to the results, and the "%" should be inserted after the precision values and says score for the sake of correctness.
It should be inserted in the Keywords: Time-of-Flight Magnetic Resonance Angiography (TOF-MRA) images.
Introduction
The section is well structured and written in good English, it introduces the problem in a broad way providing all the useful information in the following, an abuse of references without explaining the usefulness of such references, for example: in lines 36, 37, 53, 80 which are then not explained. The discussion of related work in this section is good, and identifies the various aspects of each work analysed, although it might be thought to divide the sections or at least create a subsection for related work.
Also, here the "%" should be inserted after the Dice score values for accuracy.
Materials and Methods
The section does not seem very well divided, the subsections are very unbalanced, however, with these subsections there is a proper flow of information, the use of images is clear, they are well commented through the captions.
The use of GAN is interesting and quite well explained, also with the use of the figure.
* Problem with Equation (1): it is horizontal, badly formatted, it should be formatted like Equation (2).
* The figures are very explanatory and comprehensible in their entirety, especially Figure 4. However, the figures with segmentation results might be improved by overimposing the contours onto the original images.
* Regarding automated aneurysm detection to be employed in the clinical practice, please introduce the following relevant articles on highly dependable networked systems:
- Conti, V., Militello, C., Rundo, L., & Vitabile, S. (2020). A novel bio-inspired approach for high-performance management in service-oriented networks. IEEE Transactions on Emerging Topics in Computing, 9(4), 1709-1722. DOI: 10.1109/TETC.2020.3018312
- Shi, Z., Miao, C., Schoepf, U. J., Savage, R. H., Dargis, D. M., Pan, C., ... & Zhang, L. J. (2020). A clinically applicable deep-learning model for detecting intracranial aneurysm in computed tomography angiography images. Nature communications, 11, 6090. DOI: 10.1038/s41467-020-19527-w
* Section 2.D: Class imbalanced medical image segmentation is a very hot topic. Please introduce these very recent articles:
- Ma, J., Chen, J., Ng, M., Huang, R., Li, Y., Li, C., ... & Martel, A. L. (2021). Loss odyssey in medical image segmentation. Medical Image Analysis, 102035. DOI: 10.1016/j.media.2021.102035
- Yeung, M., Sala, E., Schönlieb, C. B., & Rundo, L. (2021). Unified Focal loss: Generalising Dice and cross entropy-based losses to handle class imbalanced medical image segmentation. arXiv preprint arXiv:2102.04525.
Results
The first few lines of the results contain implementation and hardware details, this information could be included within a new section of implementation details, along with that contained in the previous section where the training parameters are described, good to have them written down for the replicability of the experiments.
* Regarding the metrics, for the dice score the internal components should be explained, so True positive, False positive and False negative.
* Figure 6: With respect to which fold are these images?
Discussion
The discussion section should be enlarged especially regarding the comparison with the state-of-the-art, but within it, the discussion of the methods implemented by the authors with which they compare the proposed system is predominant.
Conclusions
The conclusions are well written although they should be expanded to comment further on the results and what has been achieved, especially with a critical eye to the dataset. Some future developments could be identified that would allow the work to continue, in terms of methodology and dataset.
Appendix
The results in the appendix seem relevant, i.e., the use of the same method on a different dataset from the one for which this method was designed, so in my opinion they should be included in the results section in a special subsection.
References
The references are correctly numbered; however, it would only be necessary to deal with the issue within the Introduction to explain the usefulness of the references used.
Round 2
Reviewer 2 Report
I congratulate the authors on a well-executed revision and recommend the article to be accepted for publication.
Author Response
Thank you very much for reviewing our paper. We improved the manuscript with your suggestions.
Reviewer 3 Report
The Authors have mostly addressed the comments from the previous Revision Round, by providing comprehensive responses. However, some additional improvements might be considered.
The last points to address can be found in what follows.
* Figure 9: the plots should be provided with higher quality and the labels should have a bigger font to be readable.
* Section 4: The computational limitations should be better explained.
* Section 5: For future work, regarding the introduction of attention mechanisms into U-Net for class imbalance problems (such as in the case of blood vessel segmentation), the following very recent approaches should be mentioned:
- Yeung, M., Sala, E., Schönlieb, C. B., & Rundo, L. (2021). Focus U-Net: A novel dual attention-gated CNN for polyp segmentation during colonoscopy. Computers in Biology and Medicine, 104815. DOI: 10.1016/j.compbiomed.2021.104815
- Guo, C., Szemenyei, M., Yi, Y., Wang, W., Chen, B., & Fan, C. (2021, January). SA-Unet: Spatial attention U-Net for retinal vessel segmentation. In 2020 25th International Conference on Pattern Recognition (ICPR) (pp. 1236-1242). IEEE. DOI: 10.1109/ICPR48806.2021.9413346
